# Extracellular Vesicles and Membrane Protrusions in Developmental Signaling

**DOI:** 10.3390/jdb10040039

**Published:** 2022-09-21

**Authors:** Callie M. Gustafson, Laura S. Gammill

**Affiliations:** 1Department of Genetics, Cell Biology and Development, University of Minnesota, 6-160 Jackson Hall, 321 Church St SE, Minneapolis, MN 55455, USA; 2Developmental Biology Center, University of Minnesota, 6-160 Jackson Hall, 321 Church St SE, Minneapolis, MN 55455, USA

**Keywords:** extracellular vesicles, exosomes, migrasomes, membrane protrusions, embryonic development

## Abstract

During embryonic development, cells communicate with each other to determine cell fate, guide migration, and shape morphogenesis. While the relevant secreted factors and their downstream target genes have been characterized extensively, how these signals travel between embryonic cells is still emerging. Evidence is accumulating that extracellular vesicles (EVs), which are well defined in cell culture and cancer, offer a crucial means of communication in embryos. Moreover, the release and/or reception of EVs is often facilitated by fine cellular protrusions, which have a history of study in development. However, due in part to the complexities of identifying fragile nanometer-scale extracellular structures within the three-dimensional embryonic environment, the nomenclature of developmental EVs and protrusions can be ambiguous, confounding progress. In this review, we provide a robust guide to categorizing these structures in order to enable comparisons between developmental systems and stages. Then, we discuss existing evidence supporting a role for EVs and fine cellular protrusions throughout development.

## 1. Introduction

The drive to understand developing organisms has persisted for millennia [1]. In the 21st century, the parallel advancement of technology and biology has enabled embryos to be appreciated at a level that likely surpasses the ancients’ wildest imaginations. Still, endless questions remain to be answered, particularly regarding how individual cells communicate with one another to form a living being. As morphogens such as Wnt, FGF, and BMP, their intracellular signaling pathways, and downstream target genes are well defined [2,3,4], characterizing the structures that embryonic cells use to deliver these signals has become the next frontier.

Extracellular vesicles (EVs) encompass a wide variety of small, membrane-bound particles that provide paracrine, autocrine and endocrine cell signaling [5,6]. A number of comprehensive reviews offer an excellent introduction to EVs generally [7,8,9]. This review aims to provide an in-depth understanding of EVs, including the extracellular protrusions that facilitate their signaling, in the context of developmental processes.

## 2. Extracellular Vesicle Categorization

Classically, EVs are separated into three main groups: exosomes, microvesicles, and apoptotic bodies. These are defined by their size, content, and biogenesis. As the field of EV biology has expanded, many subcategories of EVs have been described that bring attention to how we classify and separate EVs and other extracellular matter [10]. Additionally, in-depth studies have revealed that EV populations are more heterogeneous than originally known [10,11,12,13,14]. In this section we aim to summarize this information into four EV categories relevant to developmental systems, including the recently discovered category, migrasomes (Table 1). Lesser known, cell-type specific EVs will be assigned into these four groups as their similarities and differences are compared. We also indicate the questions that remain to be answered on how to best characterize EVs.

### 2.1. Exosomes

Exosomes are the smallest and best-defined membrane-bound vesicles. Their roles include directed cell migration [7,23], cancer progression [9], and immune cell response [24], with more functions being revealed across a multitude of cell types each year. Exosomes are identifiable by their size, anywhere between 30 to 160 nm, though typically 100 nm on average [9]. Exosomes form as intraluminal vesicles within late endosome/multivesicular bodies (MVBs) before being exocytosed into the extracellular environment [25]. The two main mechanisms for generating exosomes are endosome sorting complexes required for transport (ESCRT)-dependent and ceramide-dependent biogenesis, with different cell types favoring one or both pathways for exosome production [26].

Recently, exosome categorization became more complicated when it was shown that exosome populations are heterogeneous. Surface proteins such as CD63, CD81, or CD9 all mark exosomes but each identify certain sub-populations [11,27]. In addition, exosomes can be further sub-divided by size, as was shown in a study separating exosomes into large exosomes (90–120 nm), small exosomes (60–80 nm), and non-membranous small particles referred to as exomeres (~35 nm) [14]. In fact, competing mechanisms for exosome formation produce different size exosomes [28]. Moreover, differences in exosome biogenesis impact function. For example, sub-populations of melanoma cell-derived exosomes have distinct RNA and proteomic profiles that produce differential gene expression in recipient cells [13]. Importantly, methods to isolate distinct exosome sub-populations are being developed [12,29].

Nomenclature redundancy also obscures exosome categorization. Over the years different fields have identified extracellularly released particles in their systems, given them names and, in a sense, constantly rediscovered exosomes. Examples of cell type-specific exosome terms include: prostasomes, released from prostate epithelial cells [30]; epididymosomes from the intraluminal space of the epididymis [31]; vexosomes derived from adeno-associated viruses [32]; dexosomes secreted from dendritic cells [33]; texosomes derived from tumor cells [34]; and tolerosomes derived from intestinal epithelial cells [35]. While these vesicles contain different cell-type specific markers and have unique functions, all are examples of exosomes and not distinctly new categories of EVs, yet often they are reviewed or considered as such. This further complicates understanding of EV research, particularly when these vesicles are not well characterized.

In other cases, extracellular particles have been misidentified as exosomes. A developmentally relevant example of this is argosomes, which transport morphogens to establish gradients in the *Drosophila* wing imaginal disc [36]. Originally thought to be a membrane-bound vesicle similar to exosomes, further characterization revealed that argosomes are extracellularly released lipoprotein particles [37] similar to lipid-particle archaesomes released from organisms of the archae family [38]. While exosome biology is better understood every day, these examples illustrate the complicated nature of exosome research and the need to properly characterize and categorize these vesicles.

### 2.2. Microvesicles

Microvesicles (MVs), also known as ectosomes, overlap in size with exosomes but can be considerably larger (100 nm to 1 µm) and are significantly different in their biogenesis [15,39,40]. Unlike exosomes, MVs emerge directly from the plasma membrane through blebbing, shedding, pearling, or scission, which also contributes to differences in the bioactive molecules that MVs contain [41]. While larger MVs are easily distinguished from exosomes, smaller MVs that fall within the 100 nm range can be difficult to differentiate unless observed blebbing from the plasma membrane. Surface markers also distinguish exosomes (e.g., LAMP1) from small MVs (e.g., BSG, SLC3A2) [42].

Adding to the confusion of cell type-specific exosome names (see above), terms have been created that that encompass MVs and other EVs. Examples include prominosomes, which are released from neuroepithelial cells carrying the Prominin-1 (CD133) protein [43], and cardiosomes, which are EVs derived from cardiomyocytes [44]. As early as the 1960s, EVs within the 500 nm to 1.5 µm range were isolated from brain tissue and called synaptosomes and myelsomes [45,46], which may include MVs and other, even larger EVs. These catch-all terms help distinguish cell-type specific functions of EVs (likely reflected in distinct cargos) that justify their names; however, this terminology obscures shared features (e.g., biogenesis pathways or means of delivery) that make mechanistic comparisons difficult.

### 2.3. Large EVs

The final EV category has long been defined as apoptotic bodies; however, large oncosomes and exophers are similar in size and biogenesis. Thus, we group these EVs together in a third category, large EVs. While significant research has gone into studying exosomes and MVs, less is known about large EVs and their role in cell communication.

#### 2.3.1. Apoptotic Bodies

While exosomes and MVs are also released from dying cells, apoptotic bodies are distinct due to their size, varying from 500 nm up to 5 µm [47]. Apoptotic bodies are generated through membrane blebbing as a cell breaks down in the final step of apoptosis [48]. Like other EVs, apoptotic bodies were originally thought to seal away and discard unwanted garbage from dying cells [17,47]; however, it has become apparent that apoptotic bodies can be used to deliver important messages to surrounding healthy cells [49]. Due to their size, apoptotic bodies are able to transport larger parcels of organelles, proteins, and genetic material [17]. While much less is known about apoptotic body functions relative to exosomes and MVs, apoptotic bodies have interesting implications in instances of developmental programmed cell death during pattern formation, differentiation, and morphogenesis [50].

#### 2.3.2. Large Oncosomes

Large oncosomes are one of the biggest large EVs, ranging in size from 1 to 10 µm in diameter and carrying distinct proteins and oncogenic materials [19]. Their biogenesis also distinguishes large oncosomes from exosomes, as they form directly from the plasma membrane, similar to apoptotic bodies and MVs. Interestingly, cancer cells also release small oncosomes (100 to 400 nm), which are functionally distinct from large oncosomes [51]. Since little characterization exists of small oncosomes, due to their size and biogenesis, they may be considered a type of tumor-derived MV [51]. Because large oncosome biogenesis is poorly understood, the main distinction between large oncosomes and other types of EVs are due to their source of origin (cancer cells) and the fact that their size (up to 10 µm) well exceeds the size ranges for apoptotic bodies or MVs [52].

#### 2.3.3. Exophers

Exophers were first identified in *C. elegans* as large, membrane-bound vesicles that carry protein aggregates during neuronal cellular stress [20,53]. Exophers can also carry damaged mitochondria to maintain cellular fitness [48,50,51]. Although mostly characterized in *C. elegans*, exophers were recently observed in mouse and human brains [54] and murine cardiomyocytes [55]. Unlike exosomes and like MVs, exophers appear to form from the plasma membrane, though they are significantly larger than MVs (~4 µm) and one of the largest EVs alongside large oncosomes [56,57].

### 2.4. Migrasomes

Migrasomes are membrane-bound vesicles that range from 0.5 µm to 2 µm [21]. Although they are large EVs based on size, they deserve their own category due to their unique biogenesis. Migrasomes appear on retraction fiber trails left behind by migrating cells (see Section 3.3.1; [21]). At retraction fiber branch points, integrins pool and adhere to the extracellular matrix (ECM) [58]. At these integrin puncta, tetraspanin (Tspan)-enriched microdomains containing Tspan 4 and 7 and cholesterol stimulate migrasome budding or bubbling [58,59].

Many markers distinguish migrasomes. Migrasomes are rich with Tspans, including the common exosome marker CD63, though only a subset including Tspan 4, 7, 18, and CD81 stimulate migrasome formation [59]. Indeed, migrasomes lack other classical exosome markers such as Alix or Tsg101, making them distinct from exosomes based on protein content as well as size [60]. Proteomic analysis has identified other migrasome-specific markers, including CPG, PIGK, NDST1 and EOGT [61], while other studies have shown that actin, SYTO14, WGA, and BODIPY TR ceramide stains are able to visualize migrasomes live in cells [15,54,60,62]. These markers will be invaluable in better characterizing this new category of EV, as there is much to learn about migrasome function.

## 3. Extracellular Protrusions: Highways for EV Transport

Signaling filopodia and EVs work together in biological systems, as detailed in a comprehensive recent review [41]. Filopodia can produce EVs through scission or pearling to aid in developmentally relevant processes such as migration, secretion, and adhesion [34,63,64]. Moreover, exosomes have been observed ‘surfing’ along protrusions [62]. While extracellular protrusions are crucial in development [63], the similarities of different membranous protrusive structures often lead developmental biologists to use terminology such as tunneling nanotubes, filopodia and retraction fibers interchangeably [62]. When combined with EV nomenclature (Table 1), inaccuracy can be further compounded. It is important to identify protrusive structures with precision so that mechanisms can be compared between model organisms and developmental stages, and any cooperation with EVs can be distinguished.

To facilitate our discussion of EVs in development, we briefly discuss three categories of filamentous structures: membrane passages, fine cellular protrusions, and adherent cellular protrusions (Table 2). Many recent, extensive reviews elaborate on the characteristics and mechanisms behind these protrusions [34,65,66,67]. For the purpose of this review, protrusions and filopodia will be used interchangeably when specific types of protrusions (i.e., membrane passages) are not defined.

### 3.1. Membrane Passages

Membrane passages create long open channels between cells. Tunneling nanotubes and intercellular bridges are in this category.

#### 3.1.1. Tunneling Nanotubes

First defined in 2004, tunneling nanotubes (TNTs) are actin-rich cellular protrusions of 50 to 700 nm diameter that span up to 20 to 100 µm long [68]. TNTs can form inter-connected bundles that create open connections between cells for the transfer of materials including cytoplasm, mitochondria, and vesicles [64]. However, defining or distinguishing TNTs is difficult as TNTs can be both open or close-ended and may include microtubules to form thicker, more stabilized extensions [63]. Depending on the cell type and circumstance, TNT extension is stimulated by the interaction of certain molecules or cellular stress [69]. TNT formation is promoted by the Rho and Ral small GTPases and Eps8, and negatively regulated by the CDC42/VASP network [69,70]. TNTs have been observed in a variety of developmental systems [63].

Recently, the possibility that TNTs facilitate EV signaling has become apparent. For example, Connexin43 is trafficked into both TNTs and EVs and is required for cell communication by either structure [71]. This is fascinating given recent work by Roberto Mayor’s group showing that Connexin43 also transcriptionally regulates development [72]. In the case of close ended TNTs, vesicles form at the tips of these protrusions before being released as EVs [73]. The question remains whether EVs and protrusions carry out redundant or combined functions, or entirely separate ones. As TNTs are also capable of transferring endocytic vesicles [71], an interesting point to consider is whether exosomes, which are released extracellularly, may also be maintained as intraluminal vesicles that are transported intracellularly.

#### 3.1.2. Intercellular Bridges

Like TNTs, intracellular bridges also form a continuous open-ended membranous connection between two cells. While TNTs and intracellular bridges are both actin and tubulin-rich and have been used interchangeably in some developmental studies, they are distinct categories due to their biogenesis [63]. Intercellular bridges form between cells of the same type due to incomplete cytokinesis, whereas TNTs form de novo and can connect cells of different origins. Intercellular bridges also typically span shorter distances (0.4 µm) and have fairly thick diameters (up to 1 µm) [63]. While their biogenesis is distinct from TNTs, they may only be identified when studies can trace their origins.

### 3.2. Fine Cellular Protrusions

While membrane passages create open channels between two cells, fine cellular protrusions are close-ended and capable of releasing vesicles from their tips. They can extend across long distances and communicate with other cells through direct interactions. Cytonemes and airinemes share these characteristics and are described below. Other fine cellular protrusions that are associated with EV release include cilia and microvilli [74,75]. However, as cilia and microvilli are well characterized and easily distinguishable from other types of long, thin cellular protrusions, they will not be discussed in depth here. Reviews on their involvement in EV communication can be found in [41,76].

#### 3.2.1. Cytonemes

Originally identified during *Drosophila* wing imaginal disc patterning, cytonemes have been considered their own class of protrusions for many years. Different from intercellular bridges or TNTs, cytonemes contain no microtubules, can reach up to 700 µm in length and 200 nm in diameter, and only exist as close-ended projections. Most commonly they function as a means of transporting and establishing developmental morphogen gradients [63], both serving as structures for EVs to travel along or to release EVs from their tips [77]. Due to the fact that TNTs have also been observed as close-ended protrusions, cytonemes and TNTs are often used interchangeably.

#### 3.2.2. Airinemes

Airnemes are another fine cellular protrusion with developmental origins. Airinemes are usually less than 1 µm in width yet capable of extending and retracting over distances greater than 100 µm. While they share similarities to cytonemes, they were first discovered in zebrafish pigment cells and unlike cytonemes, contain both actin and microtubules. Their most distinguishing feature is the vesicle-like structures that form at their tips, which can contain Delta signaling ligands [66].

### 3.3. Adherent Cellular Protrusions

While membrane passages and fine cellular protrusions can stretch across vast distances without obvious substrate connections [41], adherent cellular protrusions are extracellular structures left behind migrating cells attached to the ECM [21,67]. Distinguished by the type of EV they produce, retraction fibers and exosome deposits are included here.

#### 3.3.1. Retraction Fibers

Migrating cells leave behind a distinctive network of branching membranous structures called retraction fibers, first described in 1963 [78]. Found only at the trailing edge of migrating cells, retraction fibers are actin-rich [79] and possess cholesterol and tetraspanin-enriched microdomains [21,80]. Integrins also anchor retraction fibers to the ECM and aid in migrasome formation [80]. Since they are migration-dependent structures, speed and directionality impact retraction fiber and migrasome formation with slower, turning cells producing shorter fibers with fewer migrasomes [79].

#### 3.3.2. Exosome Deposits

While typically secreted into the extracellular space, exosomes can also form through vesiculation of adherent cellular protrusions [41,67]. This process leaves behind deposits of ECM-anchored exosomes that bear similarities to migrasomes [21]. The characteristics of the relevant adherent cellular protrusions and the mechanism that produces these exosomes, as well as any similarity to migrasome formation, is unclear.

## 4. EVs and Protrusions in Development

Although EVs are recognized to contribute to developmental signaling [81,82], most EV research has been dedicated to understanding their role in cancer biology and characterizing EVs for use in therapeutics or diagnostics [83,84]. Despite their diminutive size, EVs have been documented in a variety of developmental contexts, although it is often unclear which type of EV is being released. Similarly, while the embryonic significance of membrane protrusions such as cytonemes or tunneling nanotubes is apparent, the classifications and use of protrusion terminology in developmental systems are often vague or misconstrued. Though EVs and protrusions can function separately during signaling, the possibility of their cooperation is not always taken into consideration when researching one topic or the other. Growing evidence in developmental biology has begun to connect these two mechanisms, demonstrating the importance of protrusions and their characterization alongside EV research. Moreover, as the technology to study EVs expands, so too does the ability for developmental biologists to adapt EV protocols and apply them to developmental systems (reviewed in [85]). The following section offers examples to illustrate the role of EVs and protrusions during development.

### 4.1. Pre-Gastrulation

EVs have been implicated in the earliest stages of development. Upon fertilization in *Xenopus*, exosomes are released into the perivitelline space from the zona pellucida, spermatozoa, and cortical granule [86]. Likewise, post-fertilization zygotes and early cleavage human embryos secrete CD9-positive EVs into the perivitelline space [87,88]. In mice, the pre-implantation inner cell mass releases MVs carrying laminin and fibronectin to initiate migration of extraembryonic trophoblasts, increasing implantation efficiency [89]. Maternally derived EVs in the oviduct also promote healthy development and implantation of the blastocyst, and their absence is thought to contribute to the low success rate of assisted reproductive technologies (reviewed in [90]).

Within the early embryo, EVs and protrusions work together. The blastocoel fluid of human blastocysts contains small EVs [91]. Accordingly, *Xenopus* blastomeres extend long filopodia across the blastocoel that shed EVs from their tips which are taken up by nearby blastomeres. Meanwhile on the basolateral surface of the blastocoel, short, motile filopodia interact with and traffic EVs for transfer of morphogens or maternal ligands [86,92]. Cells of the murine blastocoel also project dynamic short and long traversing filopodia, with the latter connecting trophectoderm with distant inner cell mass cells. These filopodia contain actin, exhibit signs of vesicle release, and express FGFR2 and/or ErbB3 receptors, suggesting these protrusions are involved in signal transduction [93]. While these studies do not specify the type of EVs or fine cellular protrusions involved, it is clear that EVs and protrusions play a crucial role in post-fertilization events, mammalian implantation, and blastula/blastocyst signaling.

### 4.2. Gastrulation

EVs continue to be important during gastrulation. *Xenopus* gastrulation movements force EVs from the perivitelline space and blastocoel into the archenteron, where they are taken up by post-involution epithelial cells, though the function of this is unknown [86]. The zebrafish yolk syncytial layer produces exosomes that are apparent in the bloodstream as soon as circulation commences [94]. While this exosome release is syntenin-A-dependent and syntenin-A is required for epiboly, the role of exosomes in zebrafish gastrulation is unclear [94,95]. Zebrafish gastrula-stage mesoderm and endoderm also produce migrasomes, which are necessary for proper specification of mesodermal lineages, reflected in defective organ morphogenesis [96].

Cellular protrusions also enable cell communication during the complex embryonic germ layer rearrangements of gastrulation. In sea urchin gastrulae, primary mesenchyme, secondary mesenchyme and ectodermal cells contact one another with thin filopodia, indicating directed cell–cell communication across germ layers [97]. Analogously, gastrulating murine mesodermal cells communicate with fine cellular protrusions as they collectively migrate [98]. In zebrafish, intercellular bridges that form between pre-gastrula daughter cells are maintained through gastrulation across distant cells [99]. Meanwhile, the enveloping layer produces fine actin-based protrusions during epiboly [100] and Vangl2-dependent filopodia enable dorsal convergent mesodermal cell migration [101]. Overall, gastrula stage cellular protrusions appear to provide spatiotemporal information to coordinate cell positions and movements; however, it is unknown whether EVs accompany these protrusions.

### 4.3. Patterning

While it was originally thought that morphogen gradients were created by diffusion and/or direct cell–cell contact, those models have changed with the discovery of EVs and fine cellular protrusions [102]. In particular, *Drosophila* has revealed a great deal about delivery of morphogens via EVs and protrusions. This includes Hedgehog transport by cytonemes in germline stem cells [103] and EGF signaling via protrusions in the mechanosensory organs of the legs [104]. Cytonemes containing specific receptors respond to different signals depending on the cell type. Examples include cytonemes that detect Decapentaplegic in eye discs, Delta-Notch in the air sac primordium, FGF in both air sac primordium and tracheal cells, and Wingless in wing imaginal discs [105,106,107]. Wing imaginal discs also release exosomes containing Hedgehog [108,109] and Wingless [110]. Although Wingless EVs do not diffuse or contribute to the wing imaginal disc Wingless gradient, they do affect local signaling [111,112]. More in depth descriptions and examples of EVs involved in morphogen transport of Hedgehog, Wnt, Notch, and BMPs in *Drosophila* are reviewed in [82].

Cytonemes or cytoneme-like protrusions have been implicated in vertebrate patterning as well. In the mouse and chick limb bud, SHH-producing cells present SHH on the surface of fine cellular protrusions which extend several cell diameters to interact with responding cells’ filopodia that contain SHH co-receptors [113]. Zebrafish epiblast cells transfer Wnt from Vangl2-dependent long cytonemes to pattern the neural plate, and while EVs were not directly studied, there is evidence that Wnt may be packaged within vesicle-like structures [114,115,116,117].

In other cases, protrusions partner with EVs to enable patterning. During zebrafish pigment stripe pattern formation, airinemes extend from unpigmented xanthoblasts to pigmented embryonic melanophores. Notch ligands carried within vesicular structures found at the tips of these airinemes activate Notch signaling to promote migration, separation, and stripe pattern formation [66]. In a related example of protrusion and EV coordination, EVs released from cilia in *C. elegans* neurons have been implicated in neuron-glia communication to pattern sensory organ morphogenesis [118]. Meanwhile, cytonemes from SHH-producing cells in *Drosophila* wing discs transport SHH-containing exosomes and MVs toward receptor cells. Although it is still unclear if exosomes travel within cytonemes, small vesicles were observed moving inside these protrusions [77]. These examples highlight the importance of considering EVs in the context of protrusions and vice versa, as well as the challenge to fully characterize their functions in complex developing embryos.

### 4.4. Migration

Migration is a crucial function of cells during development. Migrating mesenchymal cells dynamically extend filopodia to sense their environment and create new adhesive contacts [119,120]. Some of these protrusions become extended and specialized [121]. In addition to migration during gastrulation (discussed in Section 4.2), lateral plate mesoderm expressing EphB3b and hepatoblasts expressing EphrinB1 use long cellular protrusions to repel one another and coordinate directional migration during zebrafish liver bud formation [122]. Long filopodia are also used for migration-related processes throughout nervous system development (reviewed in [123]). For example, chick cranial neural crest cells exhibit both short and long protrusions to enable cell–cell communication as they migrate [124]. Strikingly, cytoplasmic transfer of photoconvertible GFP was apparent through these neural crest cell membranous passages [125]. This demonstrates a mechanism to communicate positional information between migratory cells, reminiscent of what was observed in sea urchin gastrulation [97].

Evidence for EVs in developmental cell migration is just beginning to emerge. *Dictyostelium* in particular have become a model for understanding EV signaling during migration [126]. Studies on *Dictyostelium* have demonstrated that the chemoattractant cAMP can be found within MVBs and released in EVs to promote directed cell migration [127]. Chick neural crest cells also release exosomes as they migrate, suggesting this may be a conserved form of communication between migratory cells; however, the contents of neural crest cell exosomes remain elusive [23]. Although sample size constraints hinder the collection of EVs from embryonic migratory cells, metastatic cancer cells release exosomes that influence migration direction [67,128] and invasiveness of less aggressive cells [129,130,131,132]. Cancer recapitulates development [133], supporting the idea that embryonic cells regulate migration via exosomes.

Since only migrating cells form migrasomes, it is not surprising that they are involved in developmentally relevant migration events. Zebrafish gastrulae release migrasomes carrying chemokines, which are required for Kupfer’s vesicle formation and left/right laterality [96]. Chick neural crest cells also deposit migrasomes as they migrate [23]. Interestingly, murine neutrophils release migrasomes to discard damaged mitochondria and maintain cellular homeostasis [134]. Neural crest cells are subject to high rates of oxidative stress [135], raising the possibility that this could be a function of neural crest cell migrasomes as well. While EVs were originally expected to disperse soluble gradients, migrasomes are deposited into the ECM and act as a bread crumb trail-like mechanism that cells pick up from the substrate itself. It will be interesting to consider whether substrate-bound and free floating EVs are used interchangeably or serve different functions.

### 4.5. Morphogenesis

Cells engaged in morphogenetic movements use protrusions to reach across long distances to contribute signals or even mechanical forces. During the complex cell movements of mouse neural tube closure, fine filamentous extensions similar to cytonemes emerge from cells on either neural fold to create bridges as the neural tube comes together [136]. This behavior has also been observed in fly embryos during dorsal closure as epithelial cells project microtubule-rich, finger-like protrusions that ‘zip’ the tissues together [137]. *Drosophila* also require filopodia as myoblasts and myotube muscle cells fuse, as their loss results in a failure of adhesion-based foci to form between cells [138]. In addition, the FGF gradient necessary for *Drosophila* tracheal branching morphogenesis is created by reciprocal protrusions, where wing imaginal disc cytonemes present membrane-tethered FGF that is received in a contact-dependent manner by FGF receptor-expressing air sac cytonemes [139,140]. Later, EVs are required for tracheal tube fusion [141]. It is, however, unclear whether any of these examples of morphogenetic protrusions also involve the release of EVs. In *C. elegans*, neurons and glia exchange EVs produced by cilia during sensory organ morphogenesis [118], supporting this possibility.

### 4.6. Differentiation

EVs regulate differentiation in many ways. Neurons produce EVs that stimulate neural differentiation of neural stem cells [142] and promote neural circuit development and function [143]. Meanwhile, osteoblasts generate MVs and apoptotic bodies that promote differentiation of zebrafish scale osteoclasts by activating Rankl signaling [144,145]. On the other hand, bone mesenchymal stem cells release apoptotic bodies to transfer proteins and miRNA to distant mesenchymal stem cells to regulate osteogenic differentiation in mice [146]. Another study showed that primary chick notochord cells release two distinct SHH-containing exosome populations with different miRNA and protein profiles; only SHH exosomes containing integrins activated ventral spinal cord differentiation [147]. Meanwhile, cultured mouse retinal progenitor cells transfer EVs containing mRNA, miRNA, and proteins to initiate retinal differentiation [148]. In the subventricular zone, neural stem cells release EVs that regulate microglia morphology, which feedback cytokines to affect neural differentiation [149]. These examples highlight the importance of signaling by EVs during differentiation.

Protrusions can also regulate differentiation. This is illustrated by the zebrafish spinal cord, where early differentiating spinal neurons extend basal protrusions that express Delta protein, which inhibits neurogenesis. These protrusions project over several cell diameters to preferentially interact with neural progenitors of the same subtype, leading to the spatiotemporal spacing pattern of neurons in the zebrafish spinal cord [150]. A latticework of protrusions also forms in pre-neurogenic chick and human embryonic spinal cord, but their function is unknown [151]. It is unclear what types of protrusions are involved in spinal cord differentiation, or whether EVs are also released.

EVs also modulate developmental potential. Embryonic stem cells release EVs that promote stemness and prevent differentiation [152]. As lineage is restricted, EVs continue to modulate potency versus differentiation. For example, EVs released by nervous system stem cells affect cell fate and morphogenesis, and may be leveraged therapeutically (reviewed in [153]). EVs and protrusions also work together. During brain development, EVs containing Prominin-1 (CD133) are present in the neural tube lumen, and neuroepithelial cells extend CD133-positive protrusions. While it is unclear if the EVs originate from these protrusions, because CD133 is a stem cell marker, this process may enable down-regulation of stem/progenitor properties and contribute to differentiation [43].

### 4.7. Homeostasis

Homeostasis is equally crucial in developing and adult organisms. As mentioned in Section 4.4, migrating mouse neutrophils discard damaged mitochondria in migrasomes to maintain cellular homeostasis in a process termed ‘mitocytosis’ [134]. Inversely, rather than expunging dysfunctional or dying materials, zebrafish basal epithelial cells uptake apoptotic bodies to stimulate Wnt signaling and maintain cell numbers [154].

EVs are critical for vascular homeostasis. As the embryo grows, CD63-positive exosomes released from the zebrafish yolk syncytial layer travel through the bloodstream to trophically support growth and vasculogenesis of the caudal vein plexus [94]. Later, neurons in larval zebrafish and rodent brains secrete miR-132-containing exosomes that are taken up by endothelial cells and required for brain vascular integrity [155]. It has been proposed that migrasomes may also contribute to vascular homeostasis due to known functions of migrasomes and the amount of migrasome-associated Tspans expressed within the cardiovascular system [156].

Cellular protrusions are critical for homeostatic maintenance of the stem cell niche (reviewed in [157]). This is most apparent in *Drosophila*, where loss of filopodial extensions disrupts the transfer of signaling molecules that sustain the hematopoietic stem cell niche [158]. Similarly in the *Drosophila* germline, microtubule-rich nanotubes provide Decapentaplegic that maintains germline stem cells of the testis stem cell niche [159]. Meanwhile niche support cells deliver cytoneme-based Hedgehog to neighboring somatic cells to maintain the *Drosophila* ovary stem cell niche [103].

### 4.8. Regeneration

Model organisms that regenerate offer unique opportunities to study regenerative mechanisms, especially when body parts are reproduced with the morphology and pattern normally created during development (see [160]). As in development, EVs play a role in regeneration. For example, *Hydra* secrete EVs containing Wnt signaling regulatory factors that modulate head and foot regeneration [161]. During zebrafish fin regeneration, blastema cells release EVs to communicate with other cells in the fin [162]. Meanwhile, highly regenerative newt cells secrete EVs that offer protective effects to mammalian cardiomyocytes [163].

Indeed, the ability of EVs to elicit behaviors in receiving cells holds great promise for regenerative biology (reviewed in [164]). Mesenchymal stem cell (MSC)-derived EVs in particular have garnered intense interest as a therapeutic approach (reviewed in [165]). MSC-derived MVs can promote kidney, cardiac, liver, and neural regeneration, demonstrating the multi-effectiveness of EVs in promoting repair [166]. MVs may do this by delivering mRNA and proteins that effect epigenetic reprogramming in target cells, as EVs stimulate upregulation of early pluripotency markers [167]. Exosomes and MVs also contribute to wound healing. During post-injury angiogenesis, mesenchymal stem/stromal cells, leukocytes, platelets, erythrocytes and endothelial cells all release EVs that either stimulate or inhibit angiogenesis, depending on the cell type [168].

## 5. Analyzing Extracellular Vesicles and Protrusions in Development

The developmental biologist working with model organism EVs faces many challenges. The International Society for Extracellular Vesicles has defined the minimal criteria that must be met for isolated EV characterization [5], which can be difficult to achieve when purifying EVs from limited embryonic cell types. Moreover, as EV heterogeneity is parsed apart, it will become more complicated to identify and characterize them [12]. In this section, we consider these two obstacles, isolation and characterization, in turn.

### 5.1. Isolation

Originally, EVs were isolated by filtration, sucrose density gradients, high speed ultracentrifugation, chromatography, precipitation, or a combination of these approaches [5]. While these methods have enabled great progress, drawbacks include the need for large volumes of starting sample, time-consuming protocols, and the risk of co-isolating non-EV contaminants. Although commercially available kits are easier to use, faster, and supposedly scalable to smaller volumes, these kits are expensive [169]. Microfluidic systems offer an exciting new avenue for developmental EV insolation that is designed for use with small samples, though these require specialized equipment [170,171].

As recently discovered ECM-anchored EVs, migrasome isolation protocols are still being developed. Currently, excessively large volumes of starting material are required on the scale of 20,000 zebrafish embryos [96]. Thus, while it is technically possible to isolate migrasomes from developmental organisms, there is room for improvement. Future studies will also need to establish quality controls to ensure migrasomes specifically are being enriched for further study [22].

### 5.2. Characterization

Advances in imaging enable a new era of developmental EV research. Lipid dyes, radio or metabolic labels, and genetic labeling can now be used to microscopically view EVs in vivo. Depending on the developmental model and approach, many aspects of EV biogenesis, secretion, uptake, and distribution can be visualized. A thorough summary of the various types of microscopy, labeling techniques, and developmental models has been helpfully and thoroughly reviewed elsewhere [85]. Several researchers have made the case for using zebrafish [172] or *Xenopus* [86] for their ease of genetic manipulation and imaging, while explaining the current tools available for studying EVs in these systems.

Thus far, the focus of migrasome research has been to define markers that identify migrasomes in biological systems. Tspans in particular are classical markers, but also stimulate migrasome formation to varying degrees, complicating their use [59]. Importantly, proteomic analysis has identified other proteins markers such as NDST1, PIGK, CPQ or EOGT that confirm the presence of migrasomes and will be critical to their future characterization [22].

## 6. Final Thoughts

With technologies for isolating and characterizing EVs and protrusions finally approaching a resolution compatible with embryos, research on these structures in developmental systems is imperative. While many functions of EVs have been revealed in the tumor cell culture system, understanding how EVs and protrusions function together in a normal physiological context is necessary to define how cancer cells reactivate developmental programming in vivo. In fact, it has even been questioned whether in vitro EV studies recapitulate in vivo contexts [9]. To remedy this, there is much to be learned in embryos and developing tissues that contain many cell types with complicated microenvironments.

Over the last 50+ years, EVs and protrusions have been identified in dozens of cell types and model organisms (see Section 4). In many cases, only recent work has differentiated between MVs or exosomes. The same issue is reflected in studies on cytonemes and other membrane protrusions. In addition, studying EVs or protrusions separately has offered insight without providing the whole picture. Better integrating these topics while maintaining consistent definitions will allow this field to grow more rapidly.

While there is much to be learned, the impacts of developmental EVs and protrusions are already emerging. In multicellular early embryos without circulatory systems, protrusion-delivered EVs carry signals and positional information over long distances to pattern tissue formation and developing structures. During gastrulation and differentiation, EVs and protrusions coordinate massive cell rearrangements and complex morphogenetic movements. Throughout development, EVs and protrusions allow cell communication in environments such as fluid-filled compartments or glycosaminoglycan-dense ECM, where simple diffusion is neither efficient nor effective. Meanwhile, migratory cells leave behind migrasomes and exosome deposits that create road maps for trailing cells to follow. Further characterizing the signals carried, embryonic regulation, biogenesis mechanisms, and developmental functions of EVs and protrusions will better elucidate their roles, offer important insight on the normal physiological function of EVs and protrusions, and provide a new perspective that deepens our understanding of the embryo.

## Figures and Tables

**Table 1 jdb-10-00039-t001:** Categories of extracellular vesicles (EVs).

Category	Size	Biogenesis	Markers	Developmental Processes (Detailed in Section 4)	Ref.
Exosomes	30–160 nm	ESCRT or ceramide dependent multivesicular bodies	CD63 CD9 CD81	Pre-gastrulation, gastrulation patterning, migration differentiation, homeostasis regeneration	[9]
Microvesicles	0.1–1 µm	Plasma membrane blebbing	ARF6 CD40	Pre-gastrulation, patterning differentiations, regeneration	[15,16]
Large EVs:					
*Apoptotic bodies*	0.5–5 µm	Cell fragmentation during apoptosis	Trp-BODIPY cyclic peptide	Differentiation, homeostasis	[17,18]
*Large oncosomes*	1–10 µm	Plasma membrane blebbing from tumor cells	Cytokeratin18	N/A	[19]
*Exophers*	~4 µm	Plasma membrane blebbing	none	N/A	[20]
Migrasomes	0.5–2 µm	Retraction fibers	Tspan4, 7 Integrinα5β1 NDST1	Migration, patterning morphogenesis, homeostasis	[21,22]

**Table 2 jdb-10-00039-t002:** Categories of membrane protrusions.

Category	Type	Size	Characteristics	Developmental Process (es) (Detailed in Section 4)	Ref.
Membranous Passages	*Tunneling nanotubes*	50–700 nm diameter 20–100 µm long	Actin-rich	Unclear	[64]
	*Intercellular bridges*	<1 µm diameter 0.4 um long	Actin- and microtubule-rich	Gastrulation	[63]
Fine cellular Protrusions	*Cytonemes*	200 nm diameter 700 µm long	Actin-rich	Patterning, morphogenesis, regeneration	[65]
	*Airinemes*	1 µm diameter 100 µm long	Actin- and microtubule-rich	Patterning	[66]
Adherent cellular protrusions	*Retraction* *fibers*	50 nm diameter	Tetraspanin-enriched	Migration	[22]
	*Exosome* *deposits*	80–160 nm diameter	CD63-enriched	Migration	[23,67]

## Data Availability

Not applicable.

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
