# Peer review of "Extracellular Vesicles and Membrane Protrusions in Developmental Signaling"

_jdb, 2022, doi:10.3390/jdb10040039_

Round 1

Reviewer 1 Report

The manuscript is very well written and does not need any changes/corrections to be made.

Author Response

No changes requested. Please see the attachment.

Reviewer 2 Report

The goal for the authors was to write a review at the crossroads of 3 fields of research: early development, extracellular vesicles (EVs) and extracellular protrusions.

The domain of early development, which is completely in the scope of this journal (J.Dev.Biol.), is the main goal through which the authors describe what is known in the 2 other fields. The other domains are the Extracellular Vesicles and extracellular protrusions. These two domains are both with many intricated and overlapping definitions. For this reason, clear explanations and categories are needed for researcher which are not used to these domains. 

The authors chose to describe both these domains by a highly detailed and structured catalog exhibiting a classification of the different categories of EVs and of extracellular protrusions. By this choice, this review will be of the highest interest for researcher which are not really used to these two fields.

Instead of simply citing other reviews, they chose to write each part embedded in their review. The pro is a resulting very detailed review, allowing the reader to have a good overview of each domain. The cons is that it is very heavy and really not straight forward.

In fact, both the EVs and then the protruding parts, which could be considered as an introduction, are very long compared to the main part on the early development: Part 2 on EVs: more than 3 pages, Part 3 on extracellular protrusions: more than 2 pages, Part 4 on EVs & Protrusions in early development: 4.5 pages. 

In addition, in the EVs part, even though the authors describe in detail the different categories of EVs but they don’t explain the creation of the generic term of “EVs”, which has been created quite recently (compared to the first articles on exosomes and microvesicles last century) by the International Society for Extracellar Vesicles (ISEV). This term was created in order to allow researchers to use a term for their samples which are not pure, i.e. with only one category of vesicles most of the times… I think the authors should begin the part on EVs with this explanation, and to cite the article, instead of writing “Classically, EVs are separated into three main groups: exosomes, microvesicles, and 40 apoptotic bodies”.

Moreover, in the extracellular protrusions part (3.), the authors write “It is important to consider signaling filopodia in EV studies” and they cite the very recent review of Kirsi Rilla (32) on the subject. I think they should clearly state that this subject has been already evocated in a very recent review and tell how they will treat the subject and how it will be different from this other review, with what gain? 

As a conclusion the authors wrote only a small paragraph (0.25 page) of Final thoughts. This is quite frustrating as the authors could have discussed the utility of membrane protrusions for EVs, especially in early development. It would add another dimension to this review if the authors could discuss the utility of membrane protrusions for EVs, especially in early development. What are the advantages at these early stages? With no vascularization and no fluids, membrane protrusions offer a way to deliver messages/molecules directly to the target cells.

One last point, there is 2 Tables in this review:

Table 1 on EVs: categories of EVs and their link in developmental processes (one column with Developmental processes from Part 4)

Table 2 on categories of membrane protrusions (one column with Developmental processes from Part 4)

These Tables are quite a good idea as they make the link between EVs (for Table 1) or extracellular protrusions (for Table 2) with the developmental processes.

But the authors need to choose when to cite them in the text, as for now there is no citation of the Tables in text…

In conclusion, I think that this review on the implications of both EVs and extracellular protrusions in early development is quite new and is a significant contribution to the field.

But it needs to be improved to clearly show how it is different from the existing reviews.

Detailed modifications requested:

Page 1, in 2. lines 40-41: Please explain the creation of the generic term of “EVs”, which has been created quite recently (compared to the first articles on exosomes and microvesicles last century) by the International Society for Extracellar Vesicles (ISEV) instead of “Classically, EVs are separated into three main groups: exosomes, microvesicles, and 40 apoptotic bodies”. And cite the corresponding article in J. Extracell. Vesicles.

Page 2, in 2.1 lines72-74: Please illustrate this sentence with at least one reference

Page 3, in 2.3 line 114: “Thus, we will gather them together in a third category as large EVs” instead of “Thus, we consider the third category…”

Page 3, in 2.3.1 lines 119-120: Please correct the sentence “Apoptotic bodies are generated through membrane blebbing as a cell fragments into vesicles”

Page4, in 3. Line 167: Please clearly state that this subject has been already evocated in a very recent review (32) and tell how you will treat the subject and how it will be different from this other review, with what gain?

Page 7, in 4.1 line 295: “a crucial role in post-fertilization” instead of “a crucial role post-fertilization”

Page 12, in 6.: It would add another dimension to your review if you could discuss the utility of membrane protrusions for EVs, especially in early development. What is the advantages at these early stages?

Please cite Table1 and Table 2 in the text

Reviewer 3 Report

I find this review to be elegantly written and to deal with a very rare topic in the broader landscape of extracellular vesicles.

I recommend acceptance in the present form

Author Response

No changes requested. Please see attachment.